# From Hero to Hijacker: Autophagy’s Double Life in Immune Patrols and Cancer Escape

**DOI:** 10.3390/cells15020102

**Published:** 2026-01-06

**Authors:** Flavie Garampon, Aurore Claude-Taupin

**Affiliations:** Université Paris Cité, INSERM UMR-S1151, CNRS UMR-S8253, Institut Necker Enfants Malades, 75015 Paris, France; flavie.garampon@inserm.fr

**Keywords:** autophagy, mechanobiology, shear stress, cancer, immunology, migration, diapedesis, extravasation

## Abstract

Cells are constantly exposed to mechanical forces that shape their behavior, survival, and fate. The autophagy machinery emerges as a central adaptive pathway in these processes, acting not only as a metabolic and quality control mechanism but also as a key regulator of membrane dynamics and mechanotransduction. Here, we review how mechanical stress influences autophagy initiation, autophagosome maturation, and lysosomal function across different cell types. We discuss parallels between leukocyte diapedesis and circulating tumor cell (CTC) extravasation, two processes that involve profound mechanical challenges and rely on autophagy-related pathways to maintain cell integrity and enable transendothelial migration. Special attention is given to the dual role of autophagy-related proteins (ATGs) in these contexts, ranging from cytoplasmic degradation dependent on lysosomal fusion to secretory functions. Understanding how mechanical forces modulate autophagy and ATG-dependent pathways may reveal novel insights into immune regulation, tumor dissemination, and potential therapeutic targets aimed at controlling inflammation and metastasis.

## 1. Introduction

Mechanical forces are crucial factors regulating organ development and function [1]. Within the vascular network, leukocytes continuously patrol until they are recruited to sites of inflammation. While circulating in body fluids, they experience a range of mechanical forces that profoundly influence their behavior and fate. Notably, leukocytes are exposed to distinct biophysical environments in the blood and lymphatic systems, each imposing specific mechanical constraints that shape their function [2,3]. During inflammation, they undergo a complex multistep process known as diapedesis, in which they traverse the endothelial barrier. This process involves sequential steps, including rolling, adhesion, and ultimately transmigration of leukocytes across the endothelium [4,5,6,7]. Intriguingly, recent studies have revealed that cancer cells can hijack diapedesis mechanisms during metastasis, a dynamic succession of events through which tumor cells disseminate to distant sites within the body [8,9]. Cancer cells that enter the hematogenous or lymphatic systems, known as circulating tumor cells (CTCs), are exposed to mechanical forces similar to those encountered by leukocytes during circulation and diapedesis, enabling them to cross the endothelial barrier and establish secondary tumors in distant organs.

Cells are capable of sensing their microenvironment and converting biophysical cues into biological signals, a process known as mechanotransduction. Among the pathways activated by mechanical forces, autophagy-related processes have emerged as particularly relevant [10,11]. Autophagy involves the formation of a double-membrane organelle, the autophagosome, which sequesters cytosolic contents, such as proteins, lipids or organelles. Upon fusion with lysosomes, these materials are degraded and recycled. Accumulating evidence indicates that the autophagy pathway modulates cellular adhesion and migration [12], regulates innate and adaptive immune responses [13] and contributes to cancer progression [14].

In this review, we examine the mechanistic interplay between mechanical forces and autophagy regulation in circulating leukocytes and CTCs, with a particular focus on how this relationship may drive diapedesis and promote cancer metastasis.

## 2. Mechanical Forces Encountered by Cells During Circulation and Endothelial Transmigration

### 2.1. Biophysical Properties of Blood and Lymphatic Vessels

During circulation, leukocytes and CTCs are continuously exposed to the biophysical properties of their microenvironment, including viscosity, flow velocity and pressure (Table 1). Blood viscosity is influenced by multiple factors, primarily its composition (most notably the concentration of red blood cells and plasma proteins), as well as flow velocity and temperature.

Blood flow velocity varies considerably depending on the vascular segment, ranging from approximately 0.1–12 mm/s in capillaries [15,16,17,18,19], 5–200 mm/s in veins [20,21] and 50–500 mm/s in arteries [22,23,24,25,26]. At lower flow velocities, enhanced interactions among blood components lead to increased viscosity. Consequently, blood viscosity can vary widely with flow intensity, from 5 cP (centiPoise) to as high as 60 cP [27]. For reference, the viscosity of water is equivalent to 1 cP, corresponding to 1 mPa·s.

Because blood is propelled by the rhythmic contractions of the heart, flow within the circulatory system is highly pulsatile, with variations depending on vessel type. Arterial flow exhibits strong pulsatility, with pressure oscillating between diastolic and systolic phases and averaging around 12 kPa [28]. In veins, pressure is also pulsatile but less regular, as it primarily depends on muscle contractions. Venous pressure typically ranges from 1–2 kPa when lying down and can reach up to 15 kPa in the feet when standing [28]. Capillaries maintain a relatively steady flow, with an average pressure of around 3 kPa, which is essential for efficient molecular exchange [29].

The lymphatic system differs markedly from the blood circulation in both composition and flow dynamics. Lymphatic fluid is composed primarily of interstitial fluid and immune cells, with approximately 40% of total immune cells found in the lymph, compared with only 2% in the blood [30]. Importantly, CTCs can enter the lymphatic system, where they can serve as efficient metastatic precursors [31] or use lymphatic vessels as a route to access the bloodstream and seed distant metastases [32]. Flow velocity in lymphatic capillaries is substantially lower than in the hematogenous system, typically ranging from 0.01 to 0.1 mm/s [33,34], mostly due to its high viscosity of approximately 1800 cP [34]. The pressure within human skin lymphatic capillaries averages around 0.5 kPa [35] and exhibits very low pulsatility, reflecting the passive nature of lymph propulsion in these vessels.

Although the focus of this review is on the impact of mechanical forces on leukocytes and CTCs during their circulation within blood and lymphatic vessels and their endothelial transmigration, it is important to note that cells embedded within tissues also experience flow from the interstitial fluid. This fluid exhibits a viscosity of 2–3.5 cP [36] and an exceptionally very low flow velocity, estimated at 0.1–2 µm/s [37].

### 2.2. Collisions Leading to Margination of Circulating Cells

Beyond the biophysical environment of the vessels, the forces acting directly on circulating cells themselves further shape their behavior and fate. As they circulate through body fluids, leukocytes and CTCs interact with various blood components, including red blood cells and platelets [38,39,40]. The size, shape, and stiffness of these cells give rise to emergent biophysical behaviors, one of which is margination, a process where stiffer cells, such as leukocytes and platelets, migrate toward the vessel walls (Figure 1A). During collisions, softer and deformable red blood cells undergo shape changes that displaces the stiffer cells toward the periphery. Repeated collisions, combined with reduced lift forces, help maintain these stiffer cells near the vessel wall. Platelets, due to their small size and relative stiffness, display margination dynamics similar as those of leukocytes. Interestingly, cancer cells in suspension increase their stiffness when exposed to shear stress [41], a response that correlates with enhanced metastatic potential [42,43,44,45]. CTCs exhibit margination behaviors that depends on blood flow velocity and red blood cell content [46,47]. Thus, the mechanical properties of CTCs, particularly their stiffness, may, like leukocytes, influence their spatial distribution within blood flow, facilitating interactions with the endothelial surface and with neutrophils, leading to more efficient metastasis formation [48]. The margination of CTCs can also enhance their interaction with platelets to facilitate extravasation, a critical step in the metastatic cascade [49].

### 2.3. Shear Stress Modulation of Adhesion and Endothelial Transmigration

While in circulation, suspended cells such as leukocytes rotate within the fluid, thereby exposing their membranes to shear stress. The level of shear stress experienced by cells depends on the viscosity and velocity of the biofluid, as well as the cell’s position within the vessel. Cells closer to the vessel wall experience higher shear stress, and as a cell rotates, each point on its surface undergoes oscillating shear stress, with faster oscillations occurring near the wall due to increased rotational speed [50] (Figure 1B). In typical capillary blood flow, freely circulating leukocytes experience an estimated shear stress of approximately 10 dyn/cm^2^ [50]. It should also be noted that endothelial cells, lining the vessels, are themselves continuously exposed to shear stress, the magnitude of which depends on parameters such as flow velocity and fluid viscosity. Endothelial cells typically experience shear stress levels of approximately 10–20 dyn/cm^2^ in capillaries, 1–4 dyn/cm^2^ in veins, 4–30 dyn/cm^2^ in arteries and 0.64–12 dyn/cm^2^ in lymphatic vessels, as reviewed in [9]. Circulating cells traveling through these vessels are therefore subjected to shear forces of magnitudes comparable to those experienced by endothelial cells.

Diapedesis and extravasation are preceded by the rolling and adhesion of leukocytes and CTCs on the vessel wall. For leukocytes, these interactions with endothelial cells increases the shear stress on the cell membrane by up to tenfold, reaching hundreds of dyn/cm^2^ [50]. Leukocyte rolling and adhesion under high shear forces depend on multiple factors, including the stabilization of L-selectin, which is partially regulated by shear stress [51,52]. Rolling of leukocytes also relies on actin-rich membrane projections called pseudopods, which typically retract under shear stress but are maintained in the presence of inflammatory mediators [53]. Adhesion to endothelial cells is additionally mediated by integrins, such as the integrin subunit beta 2 (ITGB2 or CD18), which can be downregulated by shear stress [54]. The dynamic regulation of pseudopods and integrins allows leukocytes to avoid unnecessary rolling or adhesion under non-inflammatory conditions, while facilitating targeted diapedesis during inflammation.

CTCs similarly exploit selectins and integrins to promote rolling and intravascular arrest. Molecules such as ITGB1, ITGB3, CD24, CD44, and MUC16 expressed on CTCs bind selectins on endothelial cells and platelets, enhancing adhesion and vascular retention [55,56,57,58]. Shear stress can induce the formation of pseudopods in CTCs, promoting adhesion to endothelial cells [59]. In addition, CTCs produce an exclusive kind of actin-rich protrusions from their cell surface, called invadopodia, which are essential for extravasation [60]. Whether invadopodia formation is regulated by mechanical forces however remains to be determined.

The extravasation of CTCs through endothelial barriers is also influenced by shear forces [44]. CTCs transmigration is predominantly paracellular [60,61], although rare cases of transcellular migration have been observed in vitro [62]. Recent studies have shown that shear stress can induce the release of large extracellular vesicles (EVs) by CTCs, which in turn reprogram endothelial cells and disrupt the endothelial barrier, hence promoting cancer cell transmigration [63]. Indeed, tumor-derived EVs can remodel the endothelium in a shear stress–dependent manner, contributing to the formation of a pre-metastatic niche [64].

Endothelial cells, which partner with leukocytes during diapedesis, are also mechanosensitive. In veins, but not capillaries, shear stress under inflammatory conditions induces the expression of intercellular adhesion molecule 1 (ICAM-1), promoting leukocyte transmigration [65]. Moreover, leukocyte binding to endothelial cells induces ICAM-1 clustering, activates the calcium channel PIEZO1, and increases membrane tension under shear stress [66]. Endothelial cells respond to CTC adhesion similarly to leukocyte adhesion: adherent CTCs induce actin clustering, activating PIEZO1 and promoting extravasation [67]. These findings underscore the dual role of mechanical forces, including shear stress and membrane tension, in both endothelial cells and leukocytes, ultimately promoting efficient endothelial transmigration.

### 2.4. Tension Impacting Cell Fate Following Endothelial Transmigration

Following stable adhesion to endothelial cells, circulating leukocytes transmigrate through vessel walls in a process dependent on shear forces [68,69,70]. Diapedesis predominantly occurs at cell–cell junctions (paracellular route), but in approximately 10% of cases, it occurs directly through the body of the endothelial cell (transcellular route) [71]. In humans, leukocyte diameters are slightly less than 10 µm [72]. When migrating through intact endothelial junctions, portions of the leukocyte are compressed into openings of around 5 µm in diameter [4,73,74]. This deformation generates significant mechanical tension (up to 1.4 nN/µm) on the neutrophil plasma membrane at sites of contact with the endothelial junctions [75] (Figure 1C). For comparison, resting plasma membrane tension is in the range of 0.01–0.2 nN/µm [76,77]. During transcellular migration, increased membrane tension activates the calcium channel PIEZO1, which enhances leukocyte antibacterial activity [75]. Thus, squeezing through narrow endothelial pores serves as a key trigger for leukocyte differentiation and functional activation during inflammation.

To date, no study has directly measured membrane tension on CTCs during extravasation. Interestingly, deletion of the mechanosensitive calcium channel PIEZO1 in CTCs does not impair extravasation efficiency [67]. Similarly, while PIEZO1 is not required for neutrophil diapedesis, its absence affects neutrophil function following transmigration [75]. Accordingly, another mechanosensitive calcium channel called TRPV4, which is overexpressed in invasive cancer cells, does not appear to directly modulate transendothelial extravasation, but rather playing a role in the formation of metastatic modules [78]. Taken together, these observations suggest that extravasation of both CTCs and leukocytes is largely independent on mechanosensitive calcium channels, but further studies are required to determine the potential role of these channels in regulating cancer cell plasticity and fate after extravasation.

## 3. Autophagy

### 3.1. Autophagy and Stress Response

The previous sections illustrated the complex interplay between mechanical forces and cellular behavior during circulation and endothelial transmigration. To respond effectively to these forces, cells engage intracellular signaling pathways to regulate adhesion, migration, survival and cell differentiation. Among these pathways, autophagy emerges as a central adaptive process.

Autophagy is an adaptive catabolic mechanism that operates at basal levels to preserve cellular homeostasis and is upregulated under stress conditions. These include nutrient or growth factors deprivation, accumulation of reactive oxygen species (ROS), DNA damage, hypoxia, infection and mechanical stress [79]. Autophagy involves the formation of a double-membrane-bound structure known as the autophagosome, which selectively or non-selectively sequesters intracellular cargos within a growing membrane cisterna termed the phagophore [80]. Upon fusion with lysosomes, autophagosomal contents are degraded and recycled to support essential cellular processes [81]. In response to stressful conditions, autophagy ensures cell survival by providing essential metabolic precursors and clearing damaged organelles [82]. In immune cells, autophagy is essential for hematopoiesis [83], the maintenance of immune lineage homeostasis, antigen presentation and pathogen clearance through a process called xenophagy [84]. In healthy epithelial cells, autophagy acts as a tumor suppressor by degrading damaged contents and reducing ROS-induced genomic instability [85]. Paradoxically, in established cancers, autophagy enables cells to withstand hypoxia and detachment from the extracellular matrix (ECM), sustaining metabolism and promoting adaptation to mechanical and microenvironmental stress [85].

### 3.2. Molecular Regulation of Autophagy

Autophagy is tightly regulated by Autophagy-related (ATG) and associated proteins that coordinate the formation and maturation of autophagosomes into autolysosomes. This dynamic process comprises several stages: initiation, nucleation and elongation of phagophores, closure and maturation of the autophagosome, each orchestrated by specific ATG complexes, as extensively reviewed by others [86,87]. At the signaling level, autophagy is primarily regulated by two upstream stress-sensing hubs: the mammalian target of rapamycin (mTOR) and AMP-activated protein kinase (AMPK) [88]. Inhibition of mTORC1 serves as a principal trigger for autophagy, by activating the ULK complex. This complex is composed of the serine/threonine kinase ULK1 or ULK2, along with ATG101, ATG13 and a 200 kDa focal adhesion kinase family-interacting protein (FIP200, also known as RB1CC1) [89,90]. The ULK complex subsequently activates the PI3KC3 complex 1 (PI3KC3-C1), which produces phosphatidyl-inositol 3 phosphate (PI3P) at a specialized region of the endoplasmic reticulum known as the omegasome, thereby initiating phagophore nucleation [91] (Figure 2). Phagophore elongation is facilitated by two ubiquitin-like conjugation systems involving the ATG12-ATG5-ATG16L1 complex and the ATG8 conjugation machinery, which mediates the lipidation of Atg8 (in yeast) or its mammalian homologs of the ATG8 protein family: LC3A, LC3B, LC3C and GABARAP, GABARAPL1 and GABARAPL2 [89]. This lipidation with phosphatidylethanolamine (PE), referred to as membrane atg8ylation, can occur on various membrane structures beyond the canonical double-membrane vesicles [92]. The incorporation of lipidated ATG8 proteins allows cargo recruitment into autophagosomes [93] and promotes phagophore expansion and closure [94,95,96]. Among the core autophagy proteins, ATG9A, the only integral membrane component, plays also a pivotal role in coordinating phagophore expansion through its scramblase activity [97,98,99] as well as phagophore closure by recruiting ESCRT (endosomal sorting complexes required for transport) proteins [100]. Indeed, phagophore closure is mediated by membrane scission events driven by the ESCRT machinery [96,101,102]. Following phagophore closure, SNARE (soluble N-ethylmaleimide-sensitive factor attachment protein receptor)-dependent processes facilitate the fusion between autophagosome and lysosomes, involving notably the recruitment of STX17 on mature autophagosomes [103,104], following to the formation of a SNARE complex composed of SNAP29 and the lysosomal protein VAMP7 or VAMP8 [105].

### 3.3. Non-Canonical Roles of ATG Proteins

Recent findings have expanded the functional landscape of ATG8 family members. Indeed, they are not only restricted to double-membrane autophagosomes but can also be recruited to single-membrane compartments such as endosomes, phagosomes, macropinosomes and plasma membrane [106], as well as to lysosomes, the endoplasmic reticulum, exocytic vesicles, and other types of substrates, such as lipid droplets [92]. This process, referred to as Conjugation of ATG8s to Single Membranes (CASM), mediates diverse cellular outcomes, including cargo degradation, secretion and membrane repair [106,107,108,109]. Mechanistically, ATG8s can conjugate not only to PE, as in canonical autophagy, but also to phosphatidylserine (PS) [110]. CASM initiation differs from the canonical pathway, as it does not strictly require mTOR inactivation or ULK complex activation, yet it still relies on most of the canonical ubiquitin-like conjugation systems [111]. Functionally, several specialized CASM-related pathways have been described: LC3-associated phagocytosis (LAP) and LC3-associated endocytosis (LANDO) facilitate the internalization and subsequent degradation or recycling of extracellular material [112], whereas LC3-associated micropinocytosis (LAM) contributes to the clearance of damaged membranes. Conversely, secretory autophagy (SA) involves the release of luminal content from ATG8-positive membranes into the extracellular milieu [113]. It is also important to note that several ATG proteins participate in cellular processes that are independent of Atg8ylation and canonical autophagy. For instance, ATG9A and ATG16L1 have been implicated in plasma membrane repair [114,115], while ATG7 and BECLIN1 contribute to the regulation of cell division [116,117]. These findings highlight that ATG proteins act as multifunctional regulators of cellular homeostasis, extending their influence far beyond degradative autophagy.

### 3.4. Autophagy Regulation by Mechanical Forces

Mechanical forces, including shear stress, plasma membrane tension and compression, have a profound impact on cellular physiology and intracellular signaling [10]. To sense and respond to these cues, cells rely on an interconnected network of mechanosensitive structures. The cytoskeleton framework, including integrins, act as key mediators of force transmission and signaling [118,119]. Plasma membrane associated structures, such as the primary cilium or microvilli [120,121,122] and intracellular organelles, including the nucleus [123,124,125], also contribute to the transduction of mechanical signals into cellular responses. ATG-dependent processes play a critical role in determining cell function and fate under mechanical stress in various cell types, as outlined in recent reviews [10,11,126]. Among the mechanosensors described modulating autophagy, the primary cilium has emerged as a central hub for sensing mechanical forces and modulating downstream mTOR and AMPK signaling, key upstream regulators of autophagy. The primary cilium also influences the expression of genes impacting the autophagy pathway, through transcriptional coactivators such as YAP/TAZ. Regulation of autophagy by the primary cilium has been demonstrated in nontumorigenic contexts, including in kidney tubular cells exposed to shear stress [127] or in chondrocytes under cyclic tensile strain [128].

Hematopoietic cells, however, lack primary cilia [129], and in cancer cells, primary cilium presence appears to depend on cancer type and invasiveness [130]. Despite this, the signaling pathways activated by mechanical cues in hematopoietic and cancer cells resemble those triggered by primary cilia in other cell types. For example, cancer cell proliferation and stemness are regulated by cell density and matrix stiffness through YAP/TAZ-dependent control of autophagy [131,132]. Mechanical compression of cancer cells enhances migration through p38/MAPK- and autophagy-dependent mechanisms [133]. In addition, mechanosensing of extracellular matrix attachment is tightly linked to autophagy regulation via an integrin–mTOR axis [134]. In the following section, we will focus on the role of the mechanical forces encountered by leukocytes and cancer cells during circulation and endothelial transmigration that could influence ATG-dependent processes.

## 4. Activation of ATG-Dependent Processes in Cells During Circulation and Endothelial Transmigration

### 4.1. Autophagy-Related Proteins in Leukocyte Migration/Diapedesis

The autophagy pathway is essential for maintaining leukocyte homeostasis and function. In hematopoietic stem cell (HSCs), autophagy reduces oxidative damage and limits proliferation, thereby preserving HSC homeostasis [135,136,137,138]. Through its degradative function, autophagy also contributes to the generation of a self-tolerant T cell repertoire [139,140] and is critical for monocyte survival and differentiation, by inhibiting apoptosis [141,142,143]. Autophagy-related processes are increasingly recognized as regulators of leukocyte recruitment during inflammation, depending on the context (Table 2). For example, Atg5 deletion in the myeloid lineage of mice reduces macrophage migration to injured kidney tissue [144], while it increases monocyte infiltration during liver inflammation [145]. Similarly, the autophagy receptors NBR1 and OPTN have also been reported to decrease the migration capacities of macrophages and dendritic cells, though this effect appears independent of their canonical autophagic functions [146,147]. These observations suggest that autophagy-related proteins can modulate leukocyte migration through both autophagy-dependent and -independent function mechanisms.

#### 4.1.1. ATGs and Immune Cell Adhesion

As stated earlier in this manuscript, immune cell infiltration depends first on the adhesion capacity of immune cells to endothelial cells. In this context, the autophagy machinery seems to be a positive regulator of immune cell adhesion. Indeed, pharmacological inhibition of autophagosome-lysosome fusion with bafilomycin A1 decreases the adhesion capacity of human peripheral blood mononuclear cells and Jurkat T cells under shear stress [148]. This effect correlates with an ATG-dependent turnover of filamin, a central cytoskeletal mechanosensor that becomes damaged under PM tension. This selective degradation of filamin, referred by the authors as Chaperone-Assisted Selective Autophagy (CASA) [148], also engages the YAP/TAZ signaling pathway to stimulate de novo filamin synthesis, thereby maintaining integrin-actin and actin-actin connections, as well as the expression of adhesion-related proteins [149]. Similarly, autophagy inhibition at early stages (Class 3 PI3K inhibition) decreases the abundance of L-selectin on naive CD8 T cells [150], which promotes lymphocyte rolling on endothelial cells [151]. In this context, L-selectin is not directly degraded by autophagy; instead, mitophagy provides amino acids though mitochondrial turnover, which are then used to sustain continuous L-selectin synthesis, maintaining its membrane expression and supporting lymphocyte migration [150,152]. A non-canonical role of LC3 has also been reported in regulating the intracellular trafficking of the adhesion protein LFA1, thereby promoting lymphocyte adhesion [153]. These findings highlight that ATG-dependent processes are essential for maintaining cytoskeletal integrity and supporting the synthesis and trafficking of adhesion proteins to the leukocyte plasma membrane, thereby enhancing their rolling and adhesion on endothelial cells.

Autophagy is also induced upon monocyte adhesion to fibronectin, which is increasingly deposited on endothelial cells under shear stress [55]. In this context, autophagy promotes monocyte survival and their differentiation into macrophages [143,154]. This differentiation process relies on mTOR inhibition and AMPK activation, which trigger autophagy to degrade CD35, a monocyte marker, thereby facilitating the transition toward a macrophage identity [154]. Autophagy also mediates the degradation of cell cycle regulators, such as cyclin D1, promoting cell cycle arrest, an essential step for monocyte-to-macrophage differentiation [154].

#### 4.1.2. Autophagy-Related Proteins and Endothelial Transmigration

Following transmigration through endothelial cells, numerous genes are regulated to inhibit apoptosis and drive differentiation [155,156]. Notably, neutrophil transmigration decreases the RNA levels of several autophagy-related genes, such as ATG5, ATG7 or ATG9 [75]. As described previously, transmigration generates increased plasma membrane tension, a mechanical cue that has been shown to modulate autophagy in other systems, such as plant cells [157]. It is therefore plausible that plasma membrane tension during leukocyte diapedesis could influence the autophagy machinery, contributing to functional adaptations. Further studies are needed to clarify whether plasma membrane tension regulate processes dependent on autophagy-related proteins (canonical or non-canonical) to modulate leukocyte function during inflammation.

#### 4.1.3. ATGs in Endothelial Cells Regulate Immune Cell Adhesion

Autophagy-dependent processes also interface with endothelial cell regulation of leukocyte diapedesis. The autophagy protein Atg5 restrains neutrophil transmigration by controlling surface levels of adhesion molecules [158]. Specifically, Atg5-deficient endothelial cells exhibit elevated levels of PECAM-1 (platelet endothelial cell adhesion molecule-1) at their cell surface, enhancing neutrophil adhesion and diapedesis. Mechanistically, PECAM-1 is recycled via LC3-associated phagocytosis (LAP), a process dependent on Atg5, whereby LC3-labeled single membrane phagosomes fuse with lysosomes to degrade PECAM-1. More recently, a selective form of autophagy targeting the endoplasmic reticulum (ER), termed ER-Phagy, was shown to be upregulated in endothelial cells upon binding of apelin-13 to its receptor, angiotensin domain type 1 receptor-associated protein (APJ), hence promoting monocyte adhesion [159]. Shear stress similarly activates autophagy in endothelial cells, controlling the expression of the adhesion protein ICAM-1 [160]. Deletion of Atg5 increases ICAM-1 levels in endothelial subjected to shear stress [160], suggesting that mechanical forces, by modulating the activity of autophagy-related processes in endothelial cells, can directly influence leukocyte adhesion and transmigration.
cells-15-00102-t002_Table 2Table 2Summary of the roles of autophagy-related proteins and associated processes in leukocyte migration/diapedesis.Leukocyte FunctionAutophagy-Related Protein or Process InvolvedReferencesMigrationAtg5 deletion in the myeloid lineage (LysM-Cre) decreases macrophage migration to injured kidney tissue.[144]Atg5 deletion in the myeloid lineage (LysM-Cre) increases monocyte infiltration during liver inflammation[145]NBR1 deletion in the myeloid lineage (LysM-Cre) reduces migration during inflammation in obesity. Although NBR1 functions as an autophagy receptor, this migratory defect has not been associated with an autophagy-dependent function.[146]OPTN deletion in dendritic cells (DC, Cd11c-Cre) leads to DC maturation defects, independent on the role of OPTN in autophagy. This leads to decreased migration of DCs toward peripheral tissues, but not impacting resident DCs. [147] AdhesionBafilomycin A1 treatment decreases adhesion of Jurkat T cells and human PBMCs under shear stress.[148]ATG5 and ATG7 depletion stabilizes SYNPO2, which regulates the chaperone-assisted selective autophagy (CASA) of the actin-crosslinking protein filamin, notably involved in promoting the formation of F-actin networks [148,161]Autophagy is induced in monocytes seeded on collagen and fibronectin, essential to promote differentiation into macrophages.[143,154]VPS34 inhibition (inhibition of autophagy initiation) reduces L-Selectin abundance on CD8 T cells. L-selectin enhances lymphocyte rolling on endothelial cells, thereby facilitating their migration into secondary lymphoid organs and inflammation sites.[150,151,162]LC3 regulates the transport of the adhesion protein LFA1, promoting the adhesion of lymphocytes (non-canonical function of LC3, independent on autophagy)[153]Endothelial transmigrationNeutrophil transmigration decreases several ATG RNA levels (ATG3, ATG4, ATG9, ATG5, ATG7)[75]Cell fateBeclin1 knockdown or autophagy inhibitors (3-MA and chloroquine) decrease cell viability upon monocyte differentiation[143]


### 4.2. Autophagy-Related Processes and Metastasis

In recent years, numerous studies have implicated the autophagy pathway in regulating the metastatic behavior of tumor cells. Several in vivo studies have demonstrated that impairing autophagy in primary tumor cells reduces the formation of secondary tumors without affecting primary tumor growth [163,164,165,166]. However, recent studies have also shown that autophagy inhibition can increase metastatic burden, often through the accumulation of autophagy receptors. For example, increased levels of P62/SQSTM1 and NBR1 in several types of cancer cells promote metastasis [167,168,169,170,171]. Similarly, increased expression of SEC62, a selective receptor involved in ER-Phagy, has been linked to enhanced metastatic potential in gastric and colorectal cancers [172,173]. Beyond ER-Phagy, other forms of selective autophagy, such as mitophagy, have also been shown to influence metastatic behavior in a tissue-specific manner. Indeed, selective autophagy of mitochondria, or mitophagy, appears to promote metastasis to the lung and liver [174,175,176,177], whereas its inhibition has been reported to favor bone metastasis [178].

This underscores that autophagy’s role is context dependent, acting either a pro- or anti-metastatic factor depending on cancer type, metastatic site or specific step of the metastatic cascade (summarized in Table 3). In the following sections, we will discuss how autophagy-related processes may support metastasis, enabling invasive cancer cells to survive and adapt to multiple extreme conditions, particularly those imposed by mechanical forces.
cells-15-00102-t003_Table 3Table 3Autophagy-related proteins and metastasis.Step of the Metastatic CascadeAutophagy-Related Protein or Process InvolvedReferencesResistance to AnoikisNontumorigenic epithelial cells: ATG5, ATG6 and ATG7 deletion enhances apoptosis following detachment.[179]Hepatocellular carcinoma (HCC): ATG5 and BECLIN1 knockdown enhances apoptosis following HCC cell detachment[180]Prostate cancer: autophagy inhibitor 3-MA increase apoptosis in cells following detachment. Rapamycin (autophagy inducer) decreases cell death following detachment.[181]Ovarian cancer: chloroquine and bafilomycin A1 reduce survival following detachment.[182]Lung cancer (H1703 cells): Bafilomycin A1 increases cell death following detachment.[183]Renal Cell Carcinoma: chloroquine increases apoptosis following detachment.[184]Human breast ductal carcinoma in situ: mTOR inhibition following ECM detachment enhances anoikis resistance[134]Gastric cancer: ATG4B-dependent autophagy enhances anoikis resistance.[185]AdhesionPlatelet-released TGF-beta1 induces an autophagy-dependent expression of N-cadherin in cancer cells, hence promoting HCC metastasis[186]Autophagy promotes invadopodia formation in ovarian cancer cells[187]Endothelial transmigrationHepatocellular carcinoma: enhanced autolysosome formation through DRAM1-VAMP8 interaction promotes extravasation[188]Pancreatic cancer: mechanical compression induces autophagy to promote treatment resistance and cell survival[189]Cervical cancer (HeLa cells): compressive stress induces autophagy to promote invasion[133]High OPTN (mitophagy receptor) levels inhibits breast cancer cell extravasation[190] Autophagy, autophagy receptors and selective autophagyATG5, ATG12 and ATG7 deletion in mammary tumors increases lung metastasis[169,171,191]ATG3 knockdown increases pulmonary metastasis (mammary tumors)[192]ATG7 knockdown (but not BECLIN1) or lysosomal inhibition (hydroxychloroquine) before metastasis formation in mammary tumors decreases pulmonary metastasis[193]NSD2-dependent autophagy increases lung metastasis (breast cancer)[194]P62/SQSTM1 overexpression (autophagy receptor) promotes lung metastasis (breast cancer)[167] P62/SQSTM1 overexpression (autophagy receptor) promotes bone metastasis (lung adenocarcinoma)[168] NBR1 (autophagy receptor) accumulation promotes pulmonary metastasis of breast cancer[169]P62/SQSTM1 and NBR1 (autophagy receptors) accumulation promotes lung metastasis (lung adenocarcinoma)[170] P62/SQSTM1 and NBR1 (autophagy receptors) accumulation promotes lung metastasis (breast cancer)[171]Overexpression of FAM134B (ER-Phagy receptor) promotes metastasis[195]Upregulation of SEC62 (ER-Phagy receptor) promotes cancer metastasis [172,173]CCDC50 (lysophagy receptor) expression is correlated with increased lung metastasis (melanoma).[196] Mitophagy deficiency (ULK1 depletion) in breast cancer cells enhances bone metastasis[178]Mitophagy enhances HCC lung metastasis[174,175]Mitophagy enhances liver metastasis (colorectal cancer)[176]Mitophagy induction through DNMT1 inhibition enhances breast cancer lung metastasis[177]


#### 4.2.1. Resistance to Anoikis

When disseminating via the bloodstream or lymphatic circulation, cancer cells lose contact with the ECM for prolonged periods. In healthy adherent cells, ECM detachment induces a form of programmed cell death known as anoikis, which is crucial for maintaining tissue homeostasis and preventing inappropriate cell migration [197]. In this context, autophagy can delay the onset of anoikis, as suppression of ATG5 or ATG7 in MCF-10A epithelial cells enhances apoptosis [179]. Under physiological conditions, this transient activation of autophagy could enable cells to survive temporarily until ECM contact is reestablished. Cancer cells, however, can resist anoikis for extended periods, allowing them to survive and migrate. This resistance involves alterations in pro-apoptotic and pro-survival signaling pathways, including autophagy-related mechanisms, as demonstrated in various cancer cell types, such as hepatocellular carcinoma [180], prostate [181], ovarian [182] and lung cancer [183]. Recently, Wu et al. revealed that anoikis resistance in highly invasive clear cell Renal Cell Carcinoma is promoted by the release of free fatty acids through a selective form of autophagy, called lipophagy [184]. Lipid metabolism is central to cancer progression, as cancer cells often accumulate lipid droplets and upregulate lipogenesis [198]. By degrading stored lipids, autophagy not only provides metabolic fuel to support metastatic survival but may also facilitate subsequent stages of the metastatic cascade [199].

#### 4.2.2. ATGs in CTC Survival and Adhesion in Circulation

Following detachment from the primary tumor and entry into the bloodstream, CTCs are exposed to shear stress, a major mechanical challenge that threatens their survival, as their circulating half-time is about 30 min [200]. The ability of suspended CTCs to resist these forces appears to correlate with their metastatic potential. Indeed, aggressive circulating breast cancer cells display greater resistance to shear stress compared with non-metastatic cell lines [201]. Upon circulation in the bloodstream, CTCs can bind to platelets, which promotes tumor metastasis in part by providing TGF-β that induces an invasive phenotype in tumor cells [202]. Interestingly, platelet-derived TGF-β has recently been shown to trigger an autophagy-dependent upregulation of N-cadherin in hepatocarcinoma cells, enhancing metastatic potential [186]. N-cadherin expression is linked to increased metastatic capacities by promoting tumor cell adhesion to the endothelium [203]. These findings suggest that platelet-induced autophagy in CTCs could facilitate their endothelial adhesion through N-cadherin upregulation. This parallels the established role of autophagy in regulating the adhesion of adherent tumor cells, notably by modulating focal adhesion disassembly [163,204] and promoting the formation of invadopodia [187], the later previously shown to enhance CTC extravasation [60]. Together, these observations support an autophagy-dependent regulation of tumor cell adhesion across different cellular and mechanical contexts.

Shear stress is known to induce autophagy in various cancer types, including cervical cancer HeLa cells [205] and hepatocellular carcinoma cells (Hep3b and HEpG2) [206,207]. Although most of these studies have been performed under adherent conditions, they provide important insights into how shear stress could modulate autophagy in CTCs. They highlighted the role of plasma membrane (PM) nanodomains such as lipid rafts and membrane-anchored mechanosensors like integrins in sensing shear stress and initiating autophagy signaling [205,207]. Among PM nanodomains, caveolae, which are PM invaginations enriched in cholesterol and sphingolipids, have been shown to promote the survival of aggressive breast cancer cells exposed to shear stress in suspension [208]. The upregulation of caveolar proteins, such as caveolin-1 (CAV1), is associated with increased metastatic potential [209], notably through the induction of invadopodia formation [210], essential for endothelial adhesion under shear stress. Similarly, stimulation of autophagy correlates with increased invadopodia formation in human ovarian cancer cells [187]. In adherent cells, CAV1 has also been shown to modulate autophagy [211,212], suggesting that caveolae could contribute to the regulation of autophagy in CTCs subjected to shear stress, thereby enhancing their survival and adhesion capacities.

Caveolae protect the PM from mechanical damage by flattening in response to membrane tension [213]. Shear stress is known to cause PM damage in cancer cells cultured in suspension [214,215]. Interestingly, prostate cancer cells, unlike nontransformed prostate epithelial cells, adapt to shear stress by stiffening [41], which may help prevent PM damage. In case of PM damage, transformed cells can activate repair mechanisms. Among the processes described for PM repair [216], several ATGs have been implicated in non-canonical repair functions, including ATG9A [114] and ATG16L1 [115,217]. Together, these findings suggest that ATG-dependent pathways, likely distinct from canonical autophagy, may contribute to maintaining PM integrity and supporting CTC survival under shear stress.

#### 4.2.3. Autophagy and Cell Fate Following Cancer Cell Extravasation

Shear stress applied to suspended cancer cells increases the production of ROS, thereby promoting tumor cell extravasation both in vitro (transendothelial assay) and in vivo (zebrafish model) [44]. Impaired mitophagy can lead to ROS accumulation [218]. Interestingly, knockdown of the mitophagy receptor OPTN has been linked to enhanced extravasation of breast cancer cells [190]. Although this phenotype was not directly attributed to mitophagy, these findings suggest that shear stress could inhibit mitophagy, thereby increasing ROS production in circulating tumor cells and promoting extravasation.

Following extravasation, CTCs must adapt to their new microenvironment to successfully initiate metastatic outgrowth. In this context, enhanced autolysosome formation was recently shown to promote metastasis in a hepatocellular carcinoma model [188]. Mechanistically, DNA Damage-Regulated Autophagy Modulator 1 (DRAM1), which expression negatively correlates with patient survival in hepatocellular carcinoma, stabilizes VAMP8 at the lysosomal surface, facilitating the fusion of autophagosomes with lysosomes [219,220]. The fusion occurs in the perinuclear region [221,222], after which autolysosomes can migrate toward the cell periphery [223]. Interestingly, peripheral lysosomal positioning promotes metastasis [224]. Although this positioning does not directly affect CTC adhesion or extravasation, it markedly enhanced their post-extravasation invasiveness, notably through lysosomal secretion to induce ECM remodeling [224].

During extravasation, CTCs encounter various mechanical stresses such as local PM tension. As mentioned earlier, a local increase in PM tension has been associated with PIEZO1 activation during CTC extravasation. PIEZO1 positively regulates autophagy in different settings [225,226,227]. In pancreatic cancer cell lines, PM tension induces autophagy to promote cell survival [189], while in cervical cancer cells, PM tension-induced autophagy enhances their invasiveness [133]. Although these studies were performed on adherent cells representative of primary tumor conditions, it is tempting to hypothesize that a similar response could occur in CTCs during extravasation, where mechanosensitive ion channels, such as PIEZO1, could modulate autophagy to facilitate ECM remodeling, possibly through lysosomal secretion, as described above.

Lysosomal secretion is part of a broader system of intercellular communication between tumor cells and their microenvironment, and it plays an essential role in sustaining tumor growth and invasion. Such communication has been shown to extend from primary tumors toward distant sites, promoting the formation of pre-metastatic niches that facilitate metastasis [228]. This can be mediated by soluble factors as well as extracellular vesicles (EVs) secreted by tumor cells [9,229]. Recent studies have linked the autophagy pathway with EV secretion, showing that inhibition of autophagosome-lysosome fusion enhances EV release [230]. Prostate tumor-derived EVs have been shown to induce a switch from lysosomal degradation to secretory autophagy in osteoblasts, thereby contributing to bone remodeling and the establishment of a bone metastatic niche [231]. These observations make it tempting to hypothesize that autophagy and EV pathways may function as interconnected partners in the regulation of metastatic progression. Further studies are required to determine the precise role of autophagosome-lysosome fusion in metastatic niche remodeling and to assess whether similar mechanisms are also activated in CTCs during the extravasation step.

### 4.3. Conclusion and Perspectives

The parallels between leukocyte diapedesis and cancer cell extravasation highlight a remarkable convergence in how distinct cell types cope with mechanical constraints imposed by tissue barriers. Both leukocytes and CTCs must endure substantial mechanical stress as they circulate in body fluids, adhere to and squeeze through the endothelial layer. These processes are accompanied by rapid changes in mechanotransduction pathways. In both contexts, emerging evidence suggests that the autophagy machinery acts as a central adaptive system, maintaining cellular integrity and modulating cell fate during mechanical stress.

An important aspect that emerges from these observations is the need to distinguish between canonical and non-canonical functions of ATGs in response to mechanical stress. Canonical autophagy primarily governs the formation of autophagosomes and lysosomal degradation of intracellular cargos, providing metabolic support and quality control. In contrast, non-canonical functions of ATGs, such as secretory autophagy or plasma membrane repair, can operate independently of lysosomal fusion or complete autophagosome formation and may directly regulate membrane integrity, signaling, and intercellular communication. Both leukocytes and CTCs likely exploit these distinct ATG-dependent processes to adapt to the mechanical challenges of transmigration or extravasation. Disentangling the contributions of canonical versus non-canonical ATG pathways will therefore be essential to understand how mechanical forces influence cell survival, differentiation, and functional plasticity, and could reveal more precise therapeutic targets to modulate immune cell recruitment or limit metastatic dissemination.

## Figures and Tables

**Figure 1 cells-15-00102-f001:**
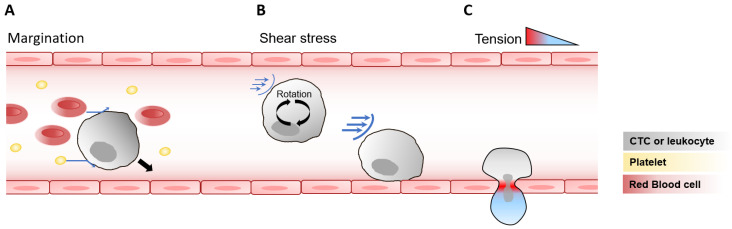
Mechanical forces acting on circulating cells and during endothelial transmigration. (**A**) *Cell–cell interactions and margination.* In blood flow, CTCs and leukocytes interact with other circulating components such as red blood cells and platelets (blue arrows), leading to displacement of stiffer cells toward the vessel wall (black arrow), a process known as margination. (**B**) *Shear stress on suspended cells*. As leukocytes and CTCs circulate in the bloodstream, they rotate within the fluid, exposing their membranes to oscillating shear stress (blue arrows), whose intensity increases closer to the vessel wall. (**C**) *Mechanical tension during endothelial transmigration*. As cells pass through the endothelial wall, they experience strong deformation and a resulting tension gradient, with maximal tension localized at endothelial junctions, as observed in leukocytes and potentially relevant to CTCs.

**Figure 2 cells-15-00102-f002:**
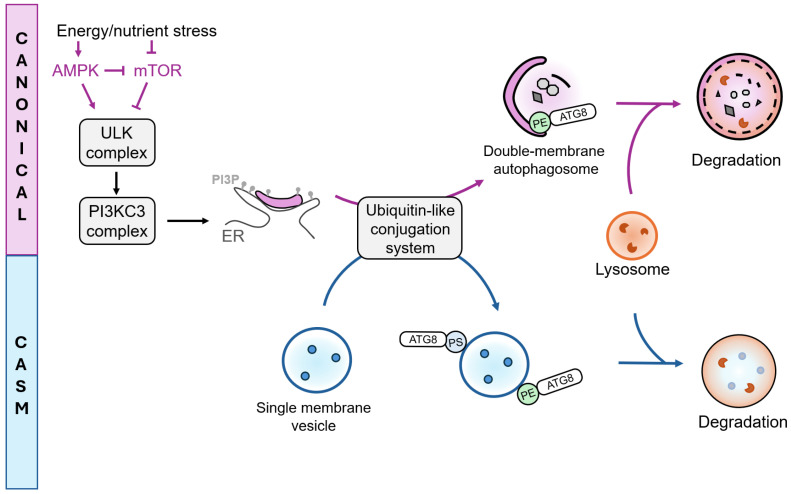
ATG-dependent processes: Canonical autophagy and CASM. Canonical autophagy and conjugation of ATG8s to Single Membranes (CASM), represent two ATG-dependent pathways distinguished by the type of membrane substrates they target. Canonical autophagy involves the formation of double-membrane autophagosomes, initiated by upstream regulation through mTOR and AMPK. Inhibition of mTOR activates the ULK and PI3KC3 complexes, leading to the production of PI3P at the endoplasmic reticulum (ER). Ubiquitin-like conjugation systems mediate ATG8 conjugation to phosphatidylethanolamine (PE) on double membrane autophagosomes (canonical autophagy) or to PE and phosphatidylserine (PS) on single membranes (CASM). Both pathways can ultimately direct cargo toward lysosomal degradation.

**Table 1 cells-15-00102-t001:** Summary of the biophysical properties of blood and lymphatic vessels.

Biophysical Property	Hematogenous System			Lymphatic System
	Capillaries	Veins	Arteries	Capillaries
Flow velocity	0.1–12 mm/s	5–200 mm/s	50–500 mm/s	0.01–0.1 mm/s
Viscosity	From 5 to 60 cP			1800 cP
Pulsatility	Low pulsatility—3 kPa	Depends on muscle contractions—From 1 to 15 kPa	Strong— Average of 12 kPa	Very low—Average of 0.5 kPa

## Data Availability

No new data were created or analyzed in this study.

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
