# Peer review of "From Hero to Hijacker: Autophagy’s Double Life in Immune Patrols and Cancer Escape"

_cells, 2026, doi:10.3390/cells15020102_

Round 1

Reviewer 1 Report

Comments and Suggestions for Authors

Garampon and Calude-Taupin summarise the dual role of autophagy during circulation and endothelial transmigration and the implications in cancer. This review will be relevant to the autophagy and cancer audience. Whereas the authors summarize the main papers and highlight the main points, there are some points that could make it even stronger:

  1. Although it is unexplored territory, what is known about selective autophagy in leukocyte migration (i.e. mitophagy) and add these concepts to the discussion.
  2. Section 4 would benefit from a table summarizing all the addressed points, and even a figure for one of the subsections as an example.

Author Response

We would like to express our sincere gratitude to the reviewer for his/her thoughtful and constructive feedback, which we believe has significantly improved the quality of our manuscript. Below, we provide a detailed response to each of the reviewer’s comments. All changes in the manuscript are indicated in red.

Reviewer 1

Garampon and Calude-Taupin summarise the dual role of autophagy during circulation and endothelial transmigration and the implications in cancer. This review will be relevant to the autophagy and cancer audience. Whereas the authors summarize the main papers and highlight the main points, there are some points that could make it even stronger:

  1. Although it is unexplored territory, what is known about selective autophagy in leukocyte migration (i.e. mitophagy) and add these concepts to the discussion.

Thank you for pointing this out. To date, only a limited number of studies have investigated the role of selective autophagy in leukocyte migration. One such study, already mentioned in our initial submission (Now Ref. 148), demonstrated that chaperone-assisted selective autophagy (CASA) promotes lymphocyte migration. In the revised manuscript, we now clarify in p.9 (section 4.1.1) that filamin turnover represents a form of selective autophagy that is critical for regulating leukocyte adhesion. In addition, we incorporated a recent study (Ref 150, p.9) showing that mitophagy is essential for the migration of naive CD8⁺ T cells by supporting the expression of key adhesion molecules.

  1. Section 4 would benefit from a table summarizing all the addressed points, and even a figure for one of the subsections as an example.

We considered adding a figure but determined that the most relevant aspects of Section 4 are better conveyed through structured comparison. We have now added two tables to improve clarity and accessibility of the information: Table 2, p.10 and Table 3, p.11-12) summarizing the role of autophagy-related proteins on leukocyte and CTC adhesion/extravasation. The two newly added tables were designed to offer a clearer and more exhaustive synthesis that a schematic could provide. In addition, we have included a schematic summary figure to visually illustrate the key concepts and mechanistic parallels discussed throughout the review.

Reviewer 2 Report

Comments and Suggestions for Authors

This review article summarizes recent studies on the involvement of autophagy in the extravasation of leucocytes and circulating tumor cells (CTC). The first part outlines the fundamental principles governing the behavior of leucocytes and CTCs in blood flow and during the extravasation process. The second part then discuss the potential roles of autophagy in these events.

              The previous studies are concisely summarized and the conclusions appear appropriate, and it may be helpful for readers to identify related references from this manuscript. However, the central theme; how mechanical stress influences autophagy and how autophagy is specifically involved in leucocyte/CTC behavior, is not sufficiently described at the molecular levels. Given that defect in autophagy affect multiple degradation pathway and can even lead to cell death, discussing specific substrates or autophagy receptors that modulate leucocyte or CTC and non-autophagic function of ATGs would be particularly informative. However, such aspects are only poorly described, and related studies may not yet exist. Therefore, it may be premature to address this topic in the form of a review article.

              The section providing general description of leucocytes/CTCs and of autophagy are disproportionately long compared with the second part, which addresses their relationship. The authors should consider reorganizing the manuscript to better highlight this main topic. In addition, the title should be revised, as the current one does not accurately reflect the content of this manuscript.

Minor comments

  • P5 “Taken together, these observations suggest that extravasation of both CTCs and leukocytes is largely independent on mechanosensitive calcium channels, but… “

Please describe that PIEZO1 function as a calcium channel in the section “2.3.1”.

  • P5 “Autophagy involves the formation of a double-membrane-bound structure known as the autophagosome, which selectively or non-selectively sequesters intracellular cargos within a growing phagophore.”

An explanation of phagophore would be required.

  • P5 “In immune cells, autophagy is essential for hematopoiesis [83], lineage-specific homeostasis, …”: In hematopoietic and immune system, ….?

  • P6 “This dynamic process comprises several stages: initiation, nucleation and elongation of phagophores, closure and maturation of the autophagosome, each orchestrated by specific ATG complexes, as extensively reviewed by others [86].:

Reference [86], published in 2011, should be updated. How about citing the following instead.

Yamamoto, H., Zhang, S. & Mizushima, N. Autophagy genes in biology and disease. Nature reviews. Genetics 24, 382-400 (2023). https://doi.org/10.1038/s41576-022-00562-w

  • P6 “Among the core autophagy proteins, ATG9A, the only integral membrane component, plays also a pivotal role in coordinating phagophore expansion through its scramblase activity [96,97], …”

Please cite the following.

Matoba, K., T. Kotani, A. Tsutsumi, T. Tsuji, T. Mori, D. Noshiro, Y. Sugita, N. Nomura, S. Iwata, Y. Ohsumi, et al. 2020. Atg9 is a lipid scramblase that mediates autophagosomal membrane expansion. Nat. Struct. Mol. Biol. 27:1185–1193. https://doi.org/10.1038/s41594-020-00518-w

  • P8 “Notably, leukocyte transmigration modulates the expression of several autophagy-related genes [65].”
  • P9 “In support with this, membrane tension during diapedesis enhances the bactericidal activity of leukocytes [65], ….”

According to the title of this reference [65], “neutrophil” rather than “leucocyte” is the appropriate term.

  • P9 “Deletion of Atg5 increases ICAM-1 levels, suggesting that mechanical forces by modulating the activity of autophagy-related processes in endothelial cells, can directly influence leukocyte adhesion and transmigration.”

Please add references.

  • P10 “Interestingly, prostate cancer cells, unlike nontransformed prostate epithelial cells, adapt to shear stress by stiffening (PMID: 25908902), which …”

Please insert the reference appropriately.

Author Response

We would like to express our sincere gratitude to the reviewer for his/her thoughtful and constructive feedback, which we believe has significantly improved the quality of our manuscript. Below, we provide a detailed response to each of the reviewer’s comments. All changes in the manuscript are indicated in red.

Reviewer 2

This review article summarizes recent studies on the involvement of autophagy in the extravasation of leucocytes and circulating tumor cells (CTC). The first part outlines the fundamental principles governing the behavior of leucocytes and CTCs in blood flow and during the extravasation process. The second part then discuss the potential roles of autophagy in these events.

              The previous studies are concisely summarized and the conclusions appear appropriate, and it may be helpful for readers to identify related references from this manuscript. However, the central theme; how mechanical stress influences autophagy and how autophagy is specifically involved in leucocyte/CTC behavior, is not sufficiently described at the molecular levels. Given that defect in autophagy affect multiple degradation pathway and can even lead to cell death, discussing specific substrates or autophagy receptors that modulate leucocyte or CTC and non-autophagic function of ATGs would be particularly informative. However, such aspects are only poorly described, and related studies may not yet exist. Therefore, it may be premature to address this topic in the form of a review article.

Following the reviewer’s recommendations, we now added a new paragraph (3.4. p.8), better describing how the autophagy pathway is regulated by mechanical forces, highlighting the roles of mechanosensors and signaling pathways described in the literature and referring to recent reviews that have already described the subject very well. We also significantly improved section 4 (p.8) to better describe at the molecular level how autophagy is involved in leukocyte/CTC behavior. We also now discuss the specific substrates of autophagy, the role of autophagy receptors and selective autophagy when relevant in leukocyte migration and endothelial transmigration (paragraph 4.1, p.9), as well as in cancer metastasis (paragraph 4.2, p.11 and 4.2.3, p.13). Moreover, we added 2 tables (Table 2, p.10 and Table 3, p.11-12) summarizing respectively the role of autophagy-related proteins on leukocyte and CTC adhesion/extravasation.

              The section providing general description of leucocytes/CTCs and of autophagy are disproportionately long compared with the second part, which addresses their relationship. The authors should consider reorganizing the manuscript to better highlight this main topic. In addition, the title should be revised, as the current one does not accurately reflect the content of this manuscript.

Thanks to the reviewer’s insightful recommendations, we reorganized the following paragraphs:

  • We made a new paragraph 3.4. (p.8) describing the role of mechanical forces in regulating autophagy in different contexts.
  • We also reorganized section 4 of our manuscript to better highlight our discussion regarding the role of autophagy and mechanical forces regulating leukocyte/CTC behavior in circulation and endothelial transmigration:

4.1. Autophagy in leukocyte migration/diapedesis (p.8)

4.1.1. Autophagy modulates immune cell adhesion (p.9)

4.1.2. Autophagy modulation following endothelial transmigration (p.9)

4.1.3. Autophagy in endothelial cells regulates immune cell adhesion (p.10)

4.2. Autophagy in metastasis (p.11)

4.2.1. Resistance to anoikis (p.12)

4.2.2. Autophagy in CTC survival and adhesion in circulation (p.13)

4.2.3. Autophagy and cell fate following cancer cell extravasation (p. 13)

We also added 2 tables (Table 2, p.10 and Table 3, p.11-12) summarizing respectively the role of autophagy-related proteins on leukocyte and CTC adhesion/extravasation. We believe that our manuscript is now more equilibrated and accurately reflects the title.

 Minor comments

  • P5 “Taken together, these observations suggest that extravasation of both CTCs and leukocytes is largely independent on mechanosensitive calcium channels, but… “

Please describe that PIEZO1 function as a calcium channel in the section “2.3.1”.

 This has been done in p.5.

  • P5 “Autophagy involves the formation of a double-membrane-bound structure known as the autophagosome, which selectively or non-selectively sequesters intracellular cargos within a growing phagophore.”

An explanation of phagophore would be required.

 An explanation of phagophore has now been added in p.5 along with a recent review reference (Ref. 80).

  • P5 “In immune cells, autophagy is essential for hematopoiesis [83], lineage-specific homeostasis, …”: In hematopoietic and immune system, ….?

 We adjusted the sentence (p.6): “In immune cells, autophagy is essential for hematopoiesis [83], the maintenance of immune lineage homeostasis,…”

  • P6 “This dynamic process comprises several stages: initiation, nucleation and elongation of phagophores, closure and maturation of the autophagosome, each orchestrated by specific ATG complexes, as extensively reviewed by others [86].:

Reference [86], published in 2011, should be updated. How about citing the following instead.

Yamamoto, H., Zhang, S. & Mizushima, N. Autophagy genes in biology and disease. Nature reviews. Genetics 24, 382-400 (2023). https://doi.org/10.1038/s41576-022-00562-w

 We replaced the reference by the one suggested and also added another (Ref. 87) in p.6.

  • P6 “Among the core autophagy proteins, ATG9A, the only integral membrane component, plays also a pivotal role in coordinating phagophore expansion through its scramblase activity [96,97], …”

Please cite the following.

Matoba, K., T. Kotani, A. Tsutsumi, T. Tsuji, T. Mori, D. Noshiro, Y. Sugita, N. Nomura, S. Iwata, Y. Ohsumi, et al. 2020. Atg9 is a lipid scramblase that mediates autophagosomal membrane expansion. Nat. Struct. Mol. Biol. 27:1185–1193. https://doi.org/10.1038/s41594-020-00518-w

We added the reference suggested in p.6.

  • P8 “Notably, leukocyte transmigration modulates the expression of several autophagy-related genes [65].”
  • P9 “In support with this, membrane tension during diapedesis enhances the bactericidal activity of leukocytes [65], ….”

According to the title of this reference [65], “neutrophil” rather than “leucocyte” is the appropriate term.

 As suggested, we now replaced leucocyte by neutrophils in the manuscript when referring to this study (p.5 and p.9).

  • P9 “Deletion of Atg5 increases ICAM-1 levels, suggesting that mechanical forces by modulating the activity of autophagy-related processes in endothelial cells, can directly influence leukocyte adhesion and transmigration.”

Please add references.

 The reference has now been added (p. 10).

  • P10 “Interestingly, prostate cancer cells, unlike nontransformed prostate epithelial cells, adapt to shear stress by stiffening (PMID: 25908902), which …”

Please insert the reference appropriately.

We apologize for this. All references are now formatted in the revised manuscript.

Reviewer 3 Report

Comments and Suggestions for Authors

In the review “From Hero to Hijacker: Autophagy’s double life in Immune Patrols and Cancer Escape”, Garampon Flavie and colleagues highlight the connection between environmental influences like mechanical stress and shear forces, autophagy and diapedesis/extravasation. First they describe the physical forces faced by cells in circulation and the basis for extravasation, as well as autophagy. Then the authors draw connection and similarities between the mechanism govern these processes in immune cells and cancer cells. Overall, the review is nicely written and easy to follow. Nevertheless, the it falls short on highlighting certain connections convincingly.

Major points:

  1. The authors are not consistent in separating which situation they are describing. The text aims to highlight how mechanical forces impact extravasation of cells, via modulating autophagy. Several factors important for this process are mixed within the text which makes it hard to follow the logic.
    1. Are the mechanical forces acting on the endothelial cells, or the leukocytes/CTCs?
    2. Are the forces happening during extravasation or in circulation?
    3. Is autophagy impacting diapedesis/ extravasation or tissue adaptation/function/survival post extravasation?
    4. Are mechanical forces impacting diapedesis/ extravasation or tissue adaptation/function/survival post extravasation?

Streamlining these points and not intermixing them to draw a narrative will be important for an understanding about the highlighted connections.

  1. The authors draw connections which are not sound. EVs can influence many aspects of the tumor. Autophagy can influence EV secretion. Therefore, the observed impacts of EV can be a consequence of autophagy. This logic is very shallow. It can be certainly discussed that there might be an interplay between these systems, but it should be done more critically.   

Detailed comments: 

In section 2.1 the description of the lymphatic system seems to be unnecessary. The presented information is detailed and well presented, but since not used later in the text rather distracting from the focus of the review.

In 2.2 the authors state that the interactions with other cells in the blood will impact leukocytes and CTCs behavior and fate. This statement is not corroborated in the following section. The relative distribution of cells within the vessels is described and how their intrinsic properties like stiffness influence their flow dynamics. The direct interaction or implied signaling events happening are not further elaborated.

In section 2.3.1 the conclusion that sheer forces and tension influence diapedesis via impacting leukocytes and endothelial cells is not supported. None of the references given shows impaired leukocyte extravasation impairment, if force sensing is absent in leukocytes. Additional references need to be provided or the conclusions altered.

In section 2.3.2, the connection between calcium signaling and sheer stress is too weak to make this suggestion. Calcium signaling is essential for many cellular processes and implying that this could indicate a requirement for sheer stress for extravasation is too indirect.

In section 3.1 the authors state “shear stress and membrane tension shape leukocyte and CTC extravasation”, while ending section 2.3.2 with “Taken together, these observations suggest that extravasation of both CTCs and leukocytes is largely independent on mechanosensitive calcium channels”. It is not clear via which mechanism and whether at all, the shear stress and membrane tension on the leukocytes or CTCs promotes extravasation.

Section 4: The implication of integrins in force transmission during circulation is not obvious and well described. The references given deal with integrin's role in interacting with the ECM which differs from the situation in circulation. Integrins play an important role during extravasation and interaction with the endothelium, but do they sense shear forces during the flow?

Reference 125 and 11 is the same. The references should be double checked, since also a PMID was provided in the text, instead of a reference.

In section 4.1 combining references 65 and 141 to highlight the point that shear stress, autophagy and cell adaptation are all connected is a stretch and not fully supported by the references. Autophagy and anti-bacterial function of macrophages are well documented in Ref. 141. Induction of autophagy during extravasation, promoting bacterial killing of neutrophils is shown in Ref. 65. Since different cell types and different context are use, making this connecting is too speculative.

Section 4.4: The section describing drosophilas wound repair (ref 175) is out of context considering that the title of the abstract is Shear stress and autophagy in cancer.

Author Response

We would like to express our sincere gratitude to the reviewer for his/her thoughtful and constructive feedback, which we believe has significantly improved the quality of our manuscript. Below, we provide a detailed response to each of the reviewer’s comments. All changes in the manuscript are indicated in red.

Reviewer 3

In the review “From Hero to Hijacker: Autophagy’s double life in Immune Patrols and Cancer Escape”, Garampon Flavie and colleagues highlight the connection between environmental influences like mechanical stress and shear forces, autophagy and diapedesis/extravasation. First they describe the physical forces faced by cells in circulation and the basis for extravasation, as well as autophagy. Then the authors draw connection and similarities between the mechanism govern these processes in immune cells and cancer cells. Overall, the review is nicely written and easy to follow. Nevertheless, the it falls short on highlighting certain connections convincingly.

Major points:

  1. The authors are not consistent in separating which situation they are describing. The text aims to highlight how mechanical forces impact extravasation of cells, via modulating autophagy. Several factors important for this process are mixed within the text which makes it hard to follow the logic.
    1. Are the mechanical forces acting on the endothelial cells, or the leukocytes/CTCs?

We apologize if this point was not sufficiently clear in the original manuscript. Both circulating cells (leukocytes and CTCs) and endothelial cells experience shear stress imposed by body fluids. To better emphasize this, we have now clarified in the opening paragraph of section 2.3. (p.4) that both circulating and endothelial cells are subjected to shear forces. Even though mechanical forces impact endothelial cells, this review aims at summarizing the impact of mechanical forces on circulating cells and to make a parallel between leukocytes and CTCs. This is now clarified at the end of the introduction (p.2)

              2. Are the forces happening during extravasation or in circulation?

As shown in Figure 1 of the first version of our manuscript, shear stress is the predominant mechanical force acting on cells during circulation, whereas membrane tension becomes the major force experienced during extravasation. To better emphasize this distinction, we have reorganized section 2 of the manuscript. The revised paragraphs now more clearly outline the dominant forces encountered in each context, as outlined below:

  • 2.3. (p.4): “While in circulation, suspended cells such as leukocytes rotate within the fluid, thereby exposing their membranes to shear stress.”)
  • 2.4. (p.5): “When migrating through intact endothelial junctions, portions of the leukocyte are compressed into openings of around 5 µm in diameter [4,63,64]. This deformation generates significant mechanical tension…”)

   3. Is autophagy impacting diapedesis/ extravasation or tissue adaptation/function/survival post extravasation?

So far, autophagy-related proteins have been shown to play an important role in circulating cell adhesion to endothelial cells, a critical event preceding extravasation. This point is now emphasized in section 4.1.1. (p.9) for leukocytes and 4.2.2. (p.13) for CTCs. We also clarified at the end of section 4.1.1. (p.9)  the reciprocal relationship whereby adhesion can positively regulate the autophagy pathway to promote survival and macrophage differentiation.

In addition, following neutrophil transmigration through endothelial cells, several autophagy-related genes have been reported to be downregulated. This has now been more clearly highlighted in paragraph 4.1.2. (p.9), along with the hypothesis that changes in autophagy gene expression may influence leukocyte function after extravasation. The role of autophagy in controlling cancer cell fate following extravasation is also described in 4.2.3. (p.13).

To better highlight the impact of autophagy in these processes, we added 2 tables (Table 2, p10 and Table 3, p11-12) summarizing respectively the role of autophagy-related proteins on leukocyte and CTC adhesion/extravasation.

    4. Are mechanical forces impacting diapedesis/ extravasation or tissue adaptation/function/survival post extravasation?

As suggested by the reviewer, we now divided section 2 in paragraphs to highlight the role of mechanical forces in diapedesis/extravasation and cell function following endothelial transmigration:

  • 2.2 Collisions leading to margination of circulating cells (p.3)
  • 2.3. Shear stress modulation of adhesion and endothelial transmigration (p.4)
  • 2.4. Tension impacting cell fate following endothelial transmigration (p.5)

Streamlining these points and not intermixing them to draw a narrative will be important for an understanding about the highlighted connections.

We thank the reviewer for the suggestion. We hope that the revised organization now highlights the key points more clearly and facilitates a better understanding of the connections discussed.

     2. The authors draw connections which are not sound. EVs can influence many aspects of the tumor. Autophagy can influence EV secretion. Therefore, the observed impacts of EV can be a consequence of autophagy. This logic is very shallow. It can be certainly discussed that there might be an interplay between these systems, but it should be done more critically.   

We added in p.14 a sentence to emphasize that a potential relationship between autophagy and EV pathways is plausible, but remains hypothetical.

Detailed comments: 

In section 2.1 the description of the lymphatic system seems to be unnecessary. The presented information is detailed and well presented, but since not used later in the text rather distracting from the focus of the review.

We chose to retain the description, as the lymphatic system contains the majority of immune cells compared to the blood circulation (as mentioned in the original version of the manuscript, now in p.2). It is also important to note that CTCs can enter the lymphatic system. We clarified this in p.2: “Importantly, CTCs can enter the lymphatic system, where they can serve as efficient metastatic precursors [31] or use lymphatic vessels as a route to access the bloodstream and seed distant metastases [32].”

In 2.2 the authors state that the interactions with other cells in the blood will impact leukocytes and CTCs behavior and fate. This statement is not corroborated in the following section. The relative distribution of cells within the vessels is described and how their intrinsic properties like stiffness influence their flow dynamics. The direct interaction or implied signaling events happening are not further elaborated.

We thank the reviewer for this comment. In the revised manuscript, we now emphasize that CTCs can form clusters with neutrophils, leading to more efficient metastasis formation (section 2.2, p.3). Moreover, we describe in the first paragraph of section 4.2.2. (p.13) a study showing that platelet-derived TGFβ activates autophagy to upregulate N-cadherin in cancer cells, promoting metastasis. We also discuss the potential impact of autophagy-dependent N-cadherin upregulation in tumor cells, which could enhance their interactions with endothelial cells.

In section 2.3.1 the conclusion that sheer forces and tension influence diapedesis via impacting leukocytes and endothelial cells is not supported. None of the references given shows impaired leukocyte extravasation impairment, if force sensing is absent in leukocytes. Additional references need to be provided or the conclusions altered.

We agree with the reviewer. We now adjusted the title of the section: now “2.4. Tension impacting cell fate following endothelial transmigration” (p.5), to corroborate the conclusions.

In section 2.3.2, the connection between calcium signaling and sheer stress is too weak to make this suggestion. Calcium signaling is essential for many cellular processes and implying that this could indicate a requirement for sheer stress for extravasation is too indirect.

We removed this sentence to avoid any confusion (p.5).

In section 3.1 the authors state “shear stress and membrane tension shape leukocyte and CTC extravasation”, while ending section 2.3.2 with “Taken together, these observations suggest that extravasation of both CTCs and leukocytes is largely independent on mechanosensitive calcium channels”. It is not clear via which mechanism and whether at all, the shear stress and membrane tension on the leukocytes or CTCs promotes extravasation.

We apologize for the lack of clarity. As noted above, we have now highlighted in the manuscript that shear stress primarily affects the adhesion capacities of circulating cells, consequently affecting endothelial transmigration (section 2.3., p.4), whereas tension predominantly influences cell fate following extravasation (section 2.4., p.5).

Section 4: The implication of integrins in force transmission during circulation is not obvious and well described. The references given deal with integrin's role in interacting with the ECM which differs from the situation in circulation. Integrins play an important role during extravasation and interaction with the endothelium, but do they sense shear forces during the flow?

The reviewer is correct that there is currently no direct evidence for a role of integrins in sensing shear forces in circulating cells. However, pseudopods, which retract under shear stress but are maintained under inflammatory conditions in leukocytes (section 2.3, p.4), have been shown to retain a pool of integrins at their tips (DOI: 10.1016/j.devcel.2007.08.012). Thus, it is plausible that integrins could sense forces upon adhesion to endothelial cells and influence cell fate. We have not included this in the revised manuscript but can add it if the reviewer feels it would be useful.

Reference 125 and 11 is the same. The references should be double checked, since also a PMID was provided in the text, instead of a reference.

All references have been carefully checked and duplicate entries have been removed.

In section 4.1 combining references 65 and 141 to highlight the point that shear stress, autophagy and cell adaptation are all connected is a stretch and not fully supported by the references. Autophagy and anti-bacterial function of macrophages are well documented in Ref. 141. Induction of autophagy during extravasation, promoting bacterial killing of neutrophils is shown in Ref. 65. Since different cell types and different context are use, making this connecting is too speculative.

Section 4.4: The section describing drosophilas wound repair (ref 175) is out of context considering that the title of the abstract is Shear stress and autophagy in cancer.

We removed these sentences.

Round 2

Reviewer 2 Report

Comments and Suggestions for Authors

The authors addressed some of my concerns by including information on autophagy receptors and summarizing the relevant literature in tables. These revisions have certainly improved this manuscript. At the same time, non-autophagic functions of ATG proteins and receptors are also highlighted. Therefore, care should be taken in the use of “autophagy” particularly in the newly added subtitles and through the text.

The authors have not addressed one of my major comments, regarding revision of the title. The terms, “hero”, Hijacker” and “double life” are not explained anywhere in the text and may mislead readers. A more appropriate title would be “Autophagy-related proteins in leukocyte migration/diapedesis and cancer cell metastasis”. While this example may appear conventional, it conveys the essential information.

Author Response

The authors addressed some of my concerns by including information on autophagy receptors and summarizing the relevant literature in tables. These revisions have certainly improved this manuscript. At the same time, non-autophagic functions of ATG proteins and receptors are also highlighted. Therefore, care should be taken in the use of “autophagy” particularly in the newly added subtitles and through the text.

We thank the reviewer for acknowledging that the 1st round of revisions improved the manuscript. To address the reviewer’s concern regarding the use of the term “autophagy”, we revised the titles 4.1. (p.8); 4.1.1. (p.9); 4.1.2. (p.9); 4.1.3. (p.10); 4.2. (p.11) and 4.2.2. (p.13) accordingly.

The authors have not addressed one of my major comments, regarding revision of the title. The terms, “hero”, Hijacker” and “double life” are not explained anywhere in the text and may mislead readers. A more appropriate title would be “Autophagy-related proteins in leukocyte migration/diapedesis and cancer cell metastasis”. While this example may appear conventional, it conveys the essential information.

We decided to keep the title unchanged, as this was not raised as a concern by the other reviewers and we do not believe it is misleading for readers.

Reviewer 3 Report

Comments and Suggestions for Authors

The authors have addressed my concerns and significantly improved the manuscript. Some minor comments for clarification are below, but the overall work will provide an advancement in the field.

Minor comments:

Line 106: migration is not a good term to describe the movement within circulation and should be replaced. 

Line 138: The made comparison between shear stress experienced by endothelial cells vs circulating cells needs to be adjusted. How can the interaction between CTC and endothelium cause shear stress greater in one vs the other. This comparison is not necessary to bring the author's point across and should be left out.

Line 164: Additional references would need to be provided to support the notion that EV supports the formation of a pre-metastatic niche. Reprogramming the blood-vessels to allow for easier extravasating is not sufficient to support this claim.

Author Response

The authors have addressed my concerns and significantly improved the manuscript. Some minor comments for clarification are below, but the overall work will provide an advancement in the field.

Minor comments:

Line 106: migration is not a good term to describe the movement within circulation and should be replaced.

We modified the sentence to remove the term migration, which could indeed have been misleading for readers (p.3).

Line 138: The made comparison between shear stress experienced by endothelial cells vs circulating cells needs to be adjusted. How can the interaction between CTC and endothelium cause shear stress greater in one vs the other. This comparison is not necessary to bring the author's point across and should be left out.

The comparison has been removed from the manuscript (p. 4).

Line 164: Additional references would need to be provided to support the notion that EV supports the formation of a pre-metastatic niche. Reprogramming the blood-vessels to allow for easier extravasating is not sufficient to support this claim.

We cannot add additional references as we have already cited the studies that specifically described the effects of shear stress on EV release by CTCs and their impact on endothelial cells.